# Characteristics, demographics, and epidemiology of possible chronic cough in Sweden: A nationwide register-based cohort study

Lotta Walz[1]*, Kristoffer Illergård[2], Johannes Arpegård[2], Cristian Dorbesi[2], Henrik Johansson[3,4], Össur Ingi Emilsson[5,6]

1 MSD (Sweden) AB, Stockholm, Sweden, 2 Division of Research, Informatics & Visualization, Reveal AB, Stockholm, Sweden, 3 Department of Medical Sciences: Clinical Physiology, Uppsala University, Uppsala, Sweden, 4 Department of Women's and Children's Health: Physiotherapy, Uppsala University, Uppsala, Sweden, 5 Department of Medical Sciences: Respiratory, Allergy and Sleep Research, Uppsala University, Uppsala, Sweden, 6 Department of Respiratory Medicine and Allergology, Akademiska Sjukhuset, Uppsala, Sweden

* Lotta.walz@merck.com

**Data Availability Statement:** We agree with the advantages of depositing data openly in repositories. However, as our data come from

## Abstract

### Aim

To show clinical characteristics, treatments, and comorbidities in chronic cough in a nation-wide cohort.

### Methods

Two cohorts were created. A national cohort with individuals from two population-based databases; the National Patient Register and Swedish Prescribed Drug Register. Secondly, a regional cohort including primary care data. Adults with at least one cough diagnosis (ICD-10 R05) and/or individuals with ≥2 dispensed prescriptions for relevant cough-medication within the inclusion period, 2016–2018, were identified. Individuals on medications which may instigate cough or suggest acute infection or diagnosed with conditions where cough is a cardinal symptom, were excluded. Those remaining were defined as having possible refractory or unexplained chronic cough (RCC/UCC).

### Results

Altogether 62,963 individuals were identified with possible RCC/UCC, giving a national prevalence of about 1%. Mean age was 56 years and 60% were females. Many (44%) of the individuals with possible RCC/UCC visited cough relevant specialist clinics during the study period, but less than 20% received a cough diagnosis. A majority (63%) had evidence of RCC/UCC in the 10 years prior to inclusion in the study. In the regional cohort, including primary care data, the prevalence of RCC/UCC was doubled (2%). Cough medicines were mainly prescribed by primary care physicians (82%).

Swedish healthcare registers, we are not allowed to distribute the data further, and therefore cannot make our data available in repositories. On the other hand, applying for data from Swedish register keepers is open (https://www.socialstyrelsen.se/en/ and https://www.regionostergotland.se/ro), provided a Swedish ethical approval (https://etikprovningsmyndigheten.se/en/) has been granted. Further information can be provided by the corresponding author.

**Funding:** This study was funded by MSD (Sweden) AB. The funding was used by an independent part, REVEAL, for data collection, data extractions, and to aggregate and analyze data (grant/award number: not applicable). The funder had no role in study design, data collection and analysis, decision to publish, or preparation of the manuscript.

**Competing interests:** Conflicts of interest: The authors report no conflicts of interest in this work, except that LW is employed by MSD (Sweden) AB, but with no stock ownership. ÖE has received honoraria from MSD and AstraZeneca for advisory work, unrelated to this manuscript. This does not alter our adherence to PLOS ONE policies on sharing data and materials.

## Conclusion

Most individuals with possible RCC/UCC sought medical care in primary care, and had a long history of cough, with various treatments tried, indicating a substantial burden of the condition. Referrals to specialist care were very rare. The results underline the need for a structured multidisciplinary approach and future therapeutic options.

## Introduction

Chronic cough, defined as cough lasting >8 weeks, is common in the general population, but difficult to treat [1]. Between 10–38% of individuals in a respiratory outpatient clinic in the USA have chronic cough [1], and in the Swedish general population the prevalence of habitual non-productive cough is estimated to be 7–11% [2, 3]. Severe chronic cough causes a major decline in health-related quality of life, with comorbidities including incontinence, cough syncope and dysphonia, leading to social isolation and depression [4]. Also, chronic cough is associated with more sick leave and decreased ability to work [2]. In spite of this, individuals with chronic cough often report their condition as being inadequately appreciated by healthcare [4, 5].

Individuals with chronic cough also seem to seek healthcare to a varying degree [6]. Since there are no specific diagnostic or treatment protocols for chronic cough, and specialist cough clinics are rare, there is a risk for potential discrepancies in the treatments provided for chronic cough [7]. Around half of individuals with chronic cough may still have chronic cough five years after initial evaluation [8].

The terms unexplained or refractory chronic cough (UCC/RCC) are widely used to describe chronic cough that either has no identifiable cause (such as asthma, gastroesophageal reflux disease (GERD), or rhinitis), or doesn't get better despite well-conducted treatment, respectively [2, 3]. Currently, only a few medical treatments are recommended by guidelines for cough, some being potentially addictive medications such as morphine [9]. A recent register-based study from Denmark used prescription and diagnosis healthcare register data to study chronic cough among patients seeking healthcare [10]. In that study, only 3% of the adult population could be identified as having possible chronic cough (PCC), significantly lower than reported in the general population [10]. This could be caused by a lack of clinical acknowledgement of cough, or patients not seeking healthcare. A few shortcomings in the study were that diagnosis data was not available from primary care, lacked analysis of differences between care providers or counties, and the duration of cough [10]. Therefore, better knowledge on the current real-world diagnosis and treatment of individuals with chronic cough is needed, to increase the awareness and improve healthcare for individuals with chronic cough.

The aim was to describe the real-world epidemiology of chronic cough in Sweden by using national register data. More specifically, to describe clinical characteristics and demographics as well as healthcare utilization including medical treatment of a population with possible chronic cough.

## Methods

This is a descriptive nationwide observational register-based cohort study, conducted on an adult population identified with possible chronic cough in Sweden from January 2016 – December 2018.

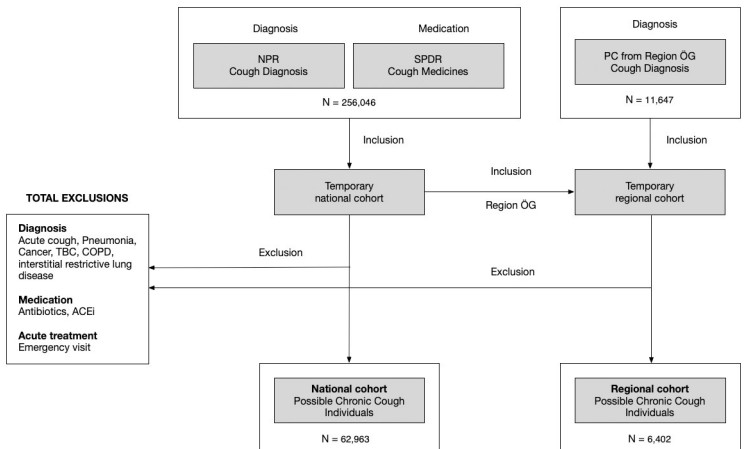

**Fig 1. Flowchart illustrating the process of identifying individuals with possible chronic cough from Swedish registers.** NPR: National Patient Register; SPDR: Swedish Prescribed Drug Register; PC: Primary Care; ÖG: Östergötland; TBC: Tuberculosis; ACEi: Angiotensin Converting Enzyme inhibitor.

Two different cohorts were created from data available in population-based registers (National Patient Register and Swedish Prescribed Drug Register) as previously described in a Danish cohort [10], with slight differences as described below (Fig 1).

*The national cohort* included individuals from two Swedish population-based databases: 1) The National Patient Register, covering information on every hospitalization or non-primary care outpatient visits of all Swedish citizens and residents with a 100% coverage. For the included patients information on diagnoses, using ICD-10 (International Classification of Disease), was identified [11]; 2) The Swedish Prescribed Drug Register, containing information on all prescribed medicines and pharmaceutical aids dispensed at Swedish pharmacies since June 2005 (100% coverage) [12].

*The regional cohort* was created by selecting individuals residing in Östergötland county, Sweden, from the national cohort and then adding individuals identified with PCC from the county's primary care data. The same inclusion and exclusion criteria were applied.

Data covering earlier R05 diagnoses, comorbidities, and dispensed prescriptions of relevant cough medicine from 2006–2015 for the individuals in the study identified were retrospectively assessed from the same registers.

Unique personal identity numbers (PIN) were used for linking individual data between the sources. Thereafter, the data were pseudonymized by replacing the PIN with a serial number. This process was performed at the National Board of Health and Welfare before the research group received access to data. As the data was pseudonymized healthcare register data only, consent was not needed from the individuals, as confirmed by the ethics committee. The study was approved by the Swedish authority for Ethical approval Review Board (Ref: 2019–06060).

## Inclusion and exclusion criteria

Inclusion criteria were: Individuals ≥18 of age at date of inclusion, with either minimum one ICD-10 diagnosis code of R05 (cough/cough with no further specification) recorded at a healthcare visit or ≥2 dispenses of relevant cough-medication (as listed in Table A in S1 File) at least 8 weeks apart within the inclusion period, 1st January 2016–31st dec 2018. This definition was used to capture chronic cough (lasting more than 8 weeks).

Individuals were excluded if they met any of the following criteria: 1) Medications which can instigate cough or suggest acute infection, i.e. ACEi or antibiotics, respectively; 2) Conditions where cough is a cardinal symptom, such as: acute cough, pneumonia, tuberculosis, chronic obstructive pulmonary disease (COPD), lung cancer and interstitial lung disease, identified by ICD-10 codes. For acute conditions such as pneumonia or acute upper airway infection and antibiotic medications, the event date needed to be within 8 weeks (before or after) of the cough diagnosis or dispensed cough medicine for the subject to be excluded. For other conditions not considered acute, such as cancer, tuberculosis, COPD, other lung diseases or treatment with ACEi, the event could take place at any time during the inclusion period 2016–2018. As the registers do not have information on smoking, smoking status could not be used in the exclusion criteria. The exclusion steps and the exclusion groups are summarized in supplementary data (Tables B and C in S1 File).

### Possible refractory /unexplained chronic cough

All eligible individuals in the final cohort were stratified into either possible refractory chronic cough (RCC) or possible unexplained chronic cough (UCC). Individuals were defined as having possible RCC if diagnosed at least once with any of the cough associated co-diagnoses (asthma, GERD, rhinitis) and/or treated with any of the drugs associated with these conditions (at least two dispenses) during the inclusion period. Otherwise, individuals were classified as having possible UCC. The number of individuals per diagnosis is provided in Table D in S1 File, and the number of individuals per associated frug is provided in Table E in S1 File.

Cough-relevant specialist clinics were identified as each clinical type that was responsible for more than 5% of the total number of R05 diagnoses in the national data set. In this cohort cough relevant specialist clinics were specialties of Internal medicine, Ear, nose and throat, Lung, Emergency, Infectious disease, or Allergy.

Mean age and standard deviation at the date of inclusion were calculated, assuming a normalized age distribution. General population data from Statistics Sweden was used to calculate prevalence.

## Results

### National cohort

During the study period, 256,046 individuals were initially identified with at least one ICD-10 diagnosis code of R05 and / or ≥2 dispenses of relevant cough-medication in the national cohort.

After applying exclusion criteria (Fig 1, Table B in S1 File), a total of 62,963 individuals were identified with PCC (corresponding to a prevalence of 787/100,000 among inhabitants of age ≥18 years), of which 80% were identified from cough medicine dispenses only (Fig 2). The mean age was 56 years, the majority were female, and 56% had possible UCC (Table 1). The individuals identified from cough medicine dispenses only were on average, 9 years older than those identified with cough diagnosis only (Table 1). Age was normally distributed, with few individuals in the oldest and youngest age groups (Fig 3 and S1 Fig). The frequency of cough medicine dispenses varied, with 37% of the individuals having four dispenses or more during the study period (Fig 4).

Some variation was seen between counties regarding the prevalence of cough medicine dispenses and cough diagnosis. Individuals with cough medicine ranged from 499/100,000 inhabitants to 819/100,000 inhabitants, and individuals with cough diagnosis ranged from 99/100,000 inhabitants to 189/100,000 inhabitants (S2 Fig).

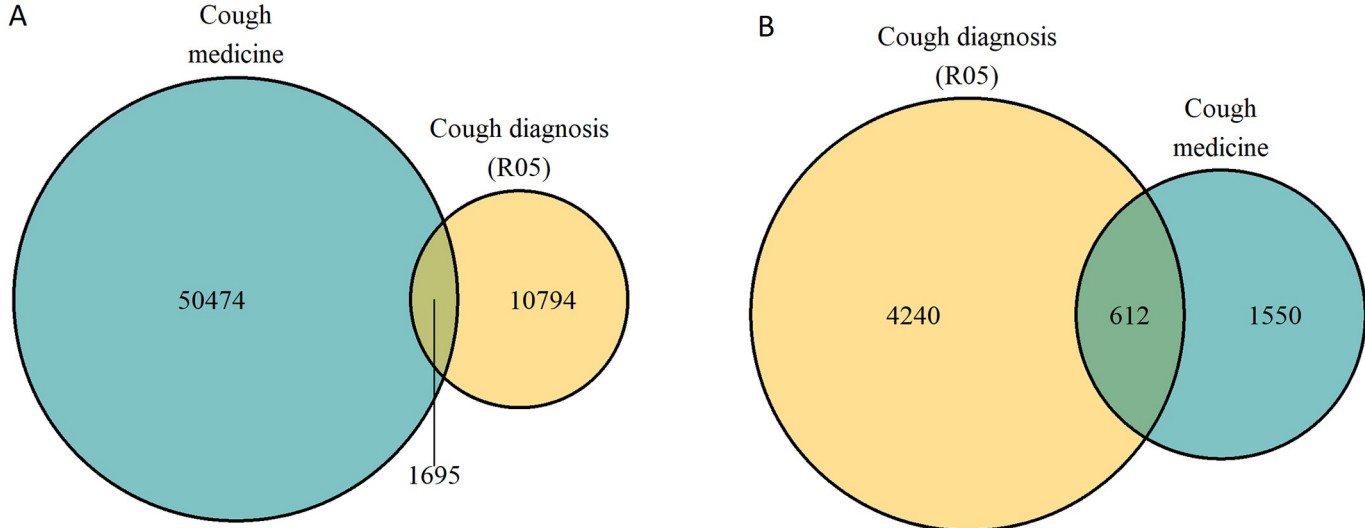

**Fig 2. Two Venn diagrams illustrating the number of individuals with possible chronic cough identified by cough medicine dispenses and/or cough diagnosis.** In turquoise individuals identified with ≥2 dispenses of relevant cough-medication in the Swedish Prescribed Drug Register, in yellow individuals with cough diagnosis (R05) a) the national cohort with cough diagnosis (R05) from the Swedish National Patient Register and b) the regional cohort with cough diagnosis (R05) from the National Patient Register and the primary care register.

Even though 44% of individuals with possible chronic cough had visited any cough relevant specialist clinic during the study period, only 41% of these individuals received a cough diagnosis code, meaning that less than 20% of the whole PCC cohort received a cough diagnosis (ICD-10 code R05, n = 12 489) (Fig 5). Of the individuals who were diagnosed with cough, the majority did not receive any other diagnosis on the same visit (72%). The most common co-diagnosis was R06 (abnormalities of breathing, 21%). Other co-diagnoses are reported in Table F in S1 File.

Cough medicines were mainly prescribed in primary care (82.5%) (Table G in S1 File). Treatment with other medications was rather common, where dispensed prescriptions for obstructive respiratory disease were most common (27,301 of those with PCC (43%)).

**Table 1. Characteristics of individuals with possible chronic cough.**

| Parameters | National cohort | | | | Regional cohort |
|---|---|---|---|---|---|
| | Cough medicine only | Cough Diagnosis only | Cough Medicine & Cough Diagnosis | Total | Total |
| Individuals (n) | 50,474 | 10,794 | 1,695 | 62,963 | 6,402 |
| Individuals (% of cohort) | 80.2% | 17.1% | 2.7% | 100% | 100% |
| Woman (%) | 60.8% | 56.6% | 61.4% | 60.1% | 58.2% |
| Age (mean (SD)) | 58 (18.0) | 49 (17.9) | 55 (16.4) | 56 (18.2) | 51 (18.0) |
| Age >65 (%) | 38.2% | 21.3% | 30.3% | 35.1% | 24.4% |
| RCC (%) | 44.1% | 41.5% | 62.2% | 44.2% | 33.3% |
| Has Asthma* (%) | 10.2% | 12.4% | 16.9% | 10.8% | 13.1% |
| Has Rhinitis* (%) | 7.1% | 12.4% | 14.6% | 8.1% | 9.0% |
| Has GERD* (%) | 5.1% | 7.9% | 14.4% | 5.8% | 8.5% |
| UCC (%) | 55.9% | 58.5% | 37.8% | 55.8% | 66.7% |

*Recorded diagnosis during 2016–2018

GERD: Gastroesophageal reflux disease; RCC: Refractory chronic cough; UCC: Unexplained chronic cough.

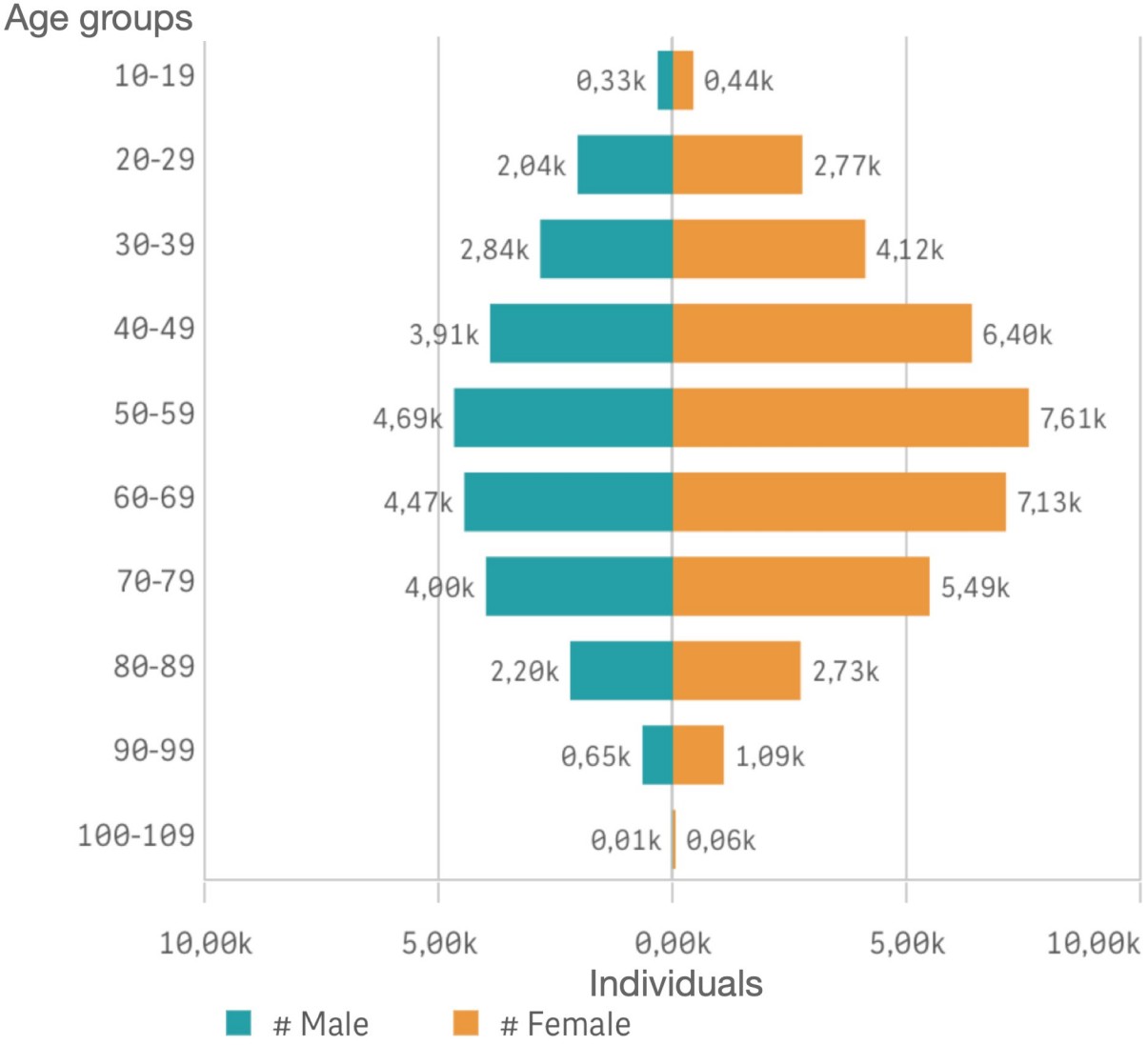

**Fig 3. Distribution of age and gender for individuals with possible chronic cough in the national cohort.**

Also, 23,300 (37%) had dispensed prescriptions of anti-acid medicine, 22,504 (36%) had dispensed prescriptions of antibiotics more than 8 weeks prior to or after the date of inclusion, 17,014 (27%) had dispensed prescriptions of systemic antihistamines, and 16,787 (27%) had dispensed prescriptions of nasal medicine. A substantial number of individuals, 5,512 (9%), had a dispensed prescription of anti-epileptic medicine (N03) but without a diagnosis of epilepsy. Further details on dispensed prescriptions are reported in Table H in S1 File.

When looking at opioids specifically, 36,838 (59%) individuals had dispensed any of the opioids (ATC code R05DA04, R05DA20 or R05FA02), and 7,653 (20%) individuals had four or more withdrawals of opioids in the three-year study period (Table I in S1 File).

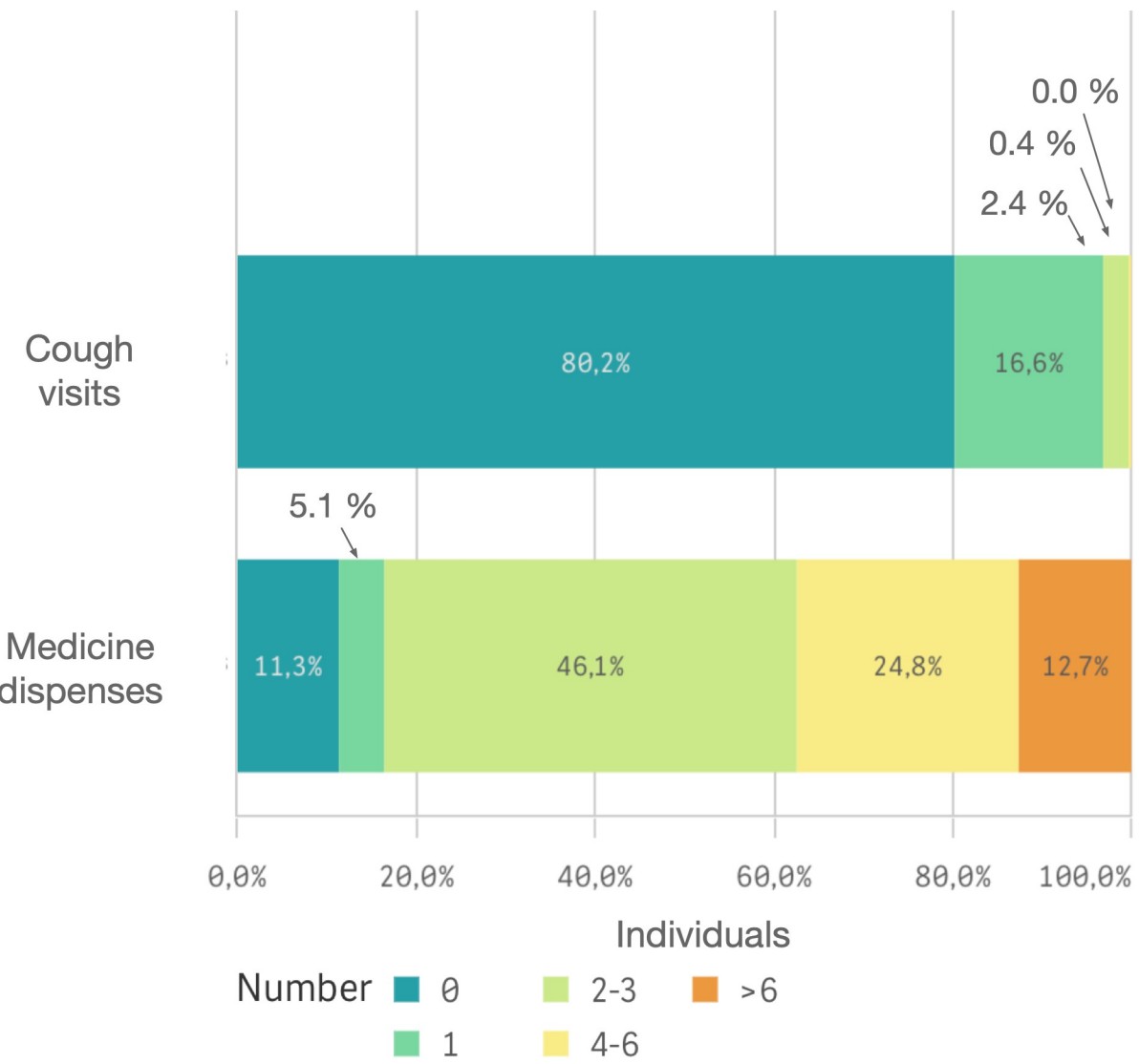

**Fig 4. Proportion of individuals with possible chronic cough stratified by number of visits with a R05 diagnosis and below proportion of individuals with possible chronic cough stratified by number of cough medicines dispensations 2016–2018.**

The national study cohort was then retrospectively analysed, where a majority (63%) of individuals with PCC in 2016–2018 had evidence of a history of cough in the 10 years-period prior to the inclusion, and most commonly already at 10 years prior to inclusion (Fig 6a). More than 30% of the individuals had used cough medicine at least 4 of the prior 10 years, and 45% had dispensed at least 2 prescriptions of cough medicine per year (Fig 6b).

### Regional cohort

In the regional cohort, where data on diagnoses from primary care was included, the prevalence of PCC was 1.8% (or 1,760 /100,000 inhabitants), mostly because of more frequent cough

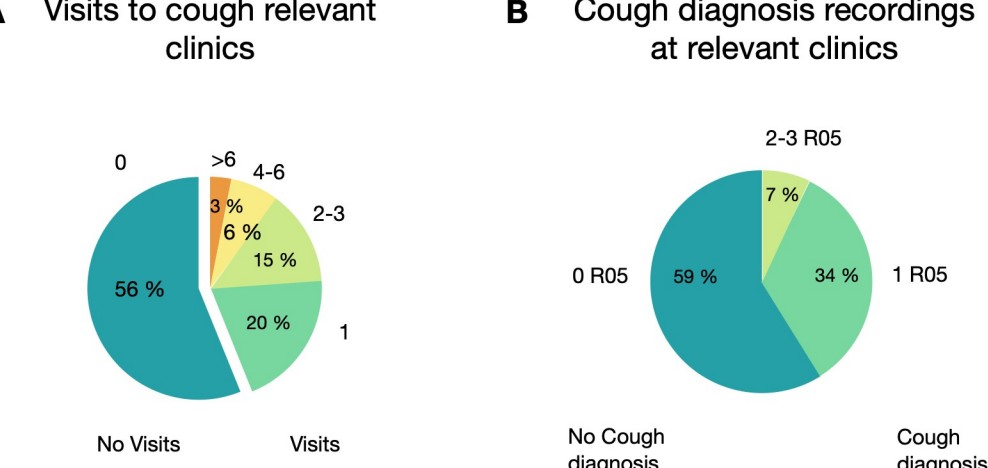

**Fig 5.** a) Individuals visiting cough relevant specialist clinics, stratified by number of visits per patient 2016–2018. b) Number of cough diagnosis recordings at cough relevant specialist clinics cough relevant clinics.

diagnoses (Fig 2b). The age and gender distribution were similar to the national cohort (Table 1). However, the prevalence in different age groups differed between the regional and national cohort. In the national cohort, the prevalence increased with increasing age groups (18–39, 40–69, ≥70 years), while in the regional cohort, the highest prevalence was seen among those 40–69 years of age (Table 2). A small proportion of individuals identified with PCC in primary care (3.5%) had a referral including a cough diagnosis to a specialist in secondary care.

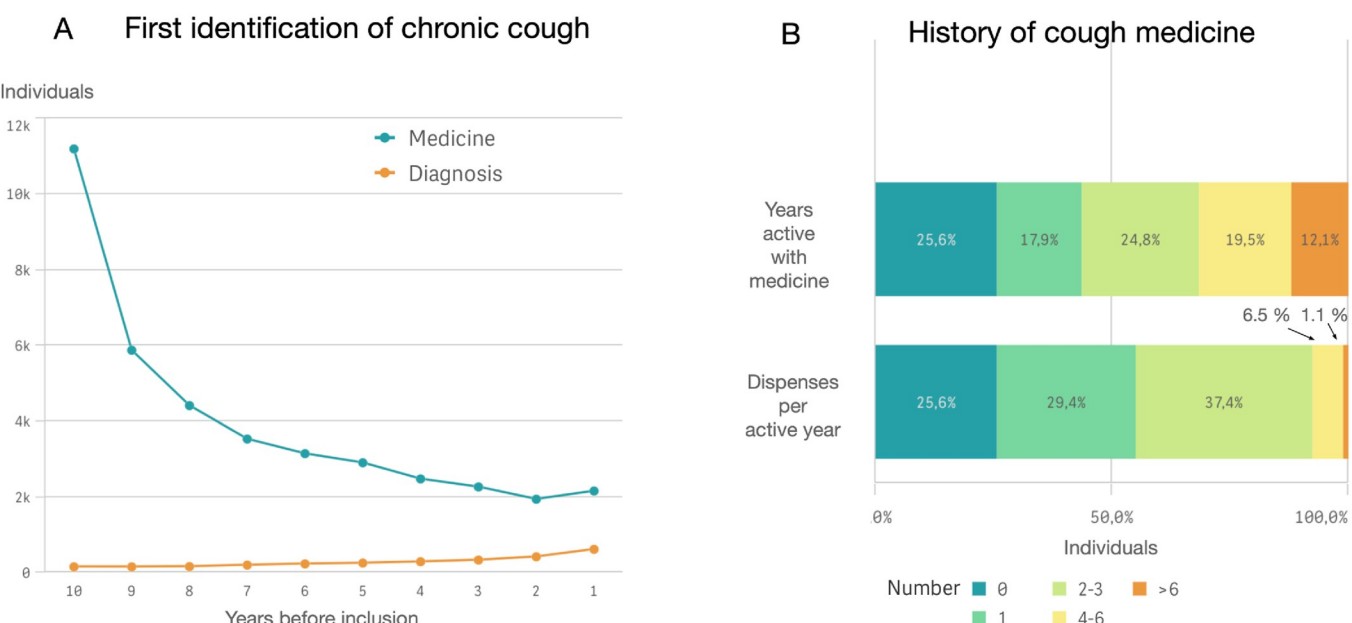

**Fig 6.** History of chronic cough a) First identification of possible chronic cough in the 10 years-period prior inclusion in the national cohort. b) Number of years 'active' and diagnoses per active year. A patient is considered to have an 'active' cough year if they have drug dispenses or cough diagnosis recordings.

**Table 2. Number of individuals in possible chronic cough cohorts and background cohorts and prevalence for different age groups.**

| Cohort | Age | PCC | Population | Prevalence |
|---|---|---|---|---|
| National | All ages | 62,963 | 80.03 K | 787 |
| | 18–39 | 12,533 | 28.53 K | 440 |
| | 40–69 | 34,190 | 36.94 K | 926 |
| | > = 70 | 16,227 | 14.58 K | 1 113 |
| Regional | All ages | 6,402 | 3.63 K | 1762 |
| | 18–39 | 1,933 | 1.31 K | 1 477 |
| | 40–69 | 3,384 | 1.65 K | 2 056 |
| | > = 70 | 1,085 | 0.68 K | 1 598 |

PCC: Possible chronic cough.

## Discussion

In this observational study, possible chronic cough (PCC) was identified in 0.8% of the Swedish adult population, using national pharmacy and secondary care registers. When primary care register data from a single healthcare region was added to the national cohort, the prevalence of PCC in that region increased to 1.8%. The highest prevalence was found in middle-aged adults, higher among females than males. Individuals with PCC were mostly managed in primary care, where also most cough medications were prescribed. Often, the cough medication included opioids. A quarter of the PCC population had two or more visits to cough-relevant clinics during the inclusion period of three years, but only a minority received a cough diagnosis. Altogether, 44% of those with PCC had an identifiable cause of chronic cough (refractory chronic cough, RCC). Most of the individuals (63%) with PCC had register-based evidence of chronic cough in the 10 year-period prior to inclusion. This supports previous findings that RCC/UCC most likely persist for years, causing substantial morbidity and impaired health status [2, 5]. A European survey on 1120 individuals with chronic cough determined that the median duration of cough was 2 to 5 years, 72% of respondents had visited their doctor more than 3 times, but only 53% had received a diagnosis [4].

We found that more than 80% of those with dispensed prescriptions of cough medication received their prescription from primary care. In line with recently published primary care data from the UK, our results support that individuals with chronic cough usually seek medical attention from a healthcare provider in primary care as the first and often only instance [13, 14]. A recent international Delphi study identified variability between geographical regions regarding the treatment of chronic cough, and a need for increased access to specialist care to improve patient care [15].

The prevalence of PCC in the present study (0.8%) was lower than in many questionnaire-based general population studies. For example, a recent epidemiological review found a prevalence of 9.6% in the general population [16]. On the other hand, studies that have used electronic healthcare records or register data have found a lower prevalence, more similar to our findings. A study using electronic healthcare records register in California, USA, found a chronic cough prevalence of 1.0% [17]. A similar study from the UK using both primary and secondary care data found a prevalence of 2.0%, without excluding individuals with COPD [13]. Possible explanations for these discrepancies in prevalence numbers may be that individuals with any of a variety of possible causes of chronic cough were excluded in the current analysis (such as COPD, treatment with ACE inhibitors, treatment with antibiotics, cancer, tuberculosis, and pneumonia), which may have led to an underestimated PCC population

[18]. Notable is that in our cohort, more than 42% of the individuals with at least one ICD-10 diagnosis code of R05 or ≥2 dispenses of relevant cough-medication were treated with ACE inhibitors.

Additionally, there is no adequate and commonly agreed method to define chronic cough in epidemiological studies [19]. A recent analysis by Hull *et al*. argued that only using structured data to identify individuals with chronic cough leads to an underestimation of the actual number since there are no approved treatments or an approved ICD-10 code for RCC/UCC (at least at that time) [13]. Also, cough is for many clinicians, regarded more as a symptom than a distinct medical condition or disease, and therefore not diagnosed and coded as such [14]. This is also supported by the present study, where individuals with possible chronic cough rarely seemed to receive a cough diagnosis on visits to cough relevant clinics.

Another possible explanation for the lower prevalence in register-based data is that individuals with chronic cough have stopped seeking healthcare for their cough, likely because of a lack of treatment modalities [9]. Therefore, these individuals cannot be identified from register data as the time period analysed is relatively short. Indeed, our retrospective data, covering a 10 years-period prior to the inclusion period, showed that the majority of individuals had evidence of PCC 10 years before the inclusion date.

The higher prevalence of PCC in females and in elderly individuals is consistent with findings in other studies [4, 19]. The most common comorbidities in the cohort of possible refractory chronic cough were gastroesophageal reflux disease (GERD), allergic rhinitis and asthma, also consistent with the current knowledge [9]. This indicates that the patient group identified with PCC using the Swedish healthcare register data is comparable to chronic cough cohorts recruited in clinical studies.

A considerable number of individuals, 36,838, of the adult population in Sweden were identified with dispensed prescriptions of opioids intended for chronic cough (i.e., around 0.5%). The prescription of opioids is of increasing concern because of addiction-related problems [20]. This is a somewhat higher proportion of opioid use compared with recent data from a study in Florida [21]. Other non-addictive treatments need to be sought for this evidently large group of patients.

Most of the individuals in this study had multiple cough medicine dispenses at least eight weeks apart, and almost half of the population had four or more dispenses during the study period, indicating significant chronic cough. In addition, 8.6% of the individuals had dispensed more than 2 prescriptions of cough-relevant anti-epileptics but with no records of associated diagnosis. Only 2% of individuals with at least four dispenses of cough medicines, had a recorded cough diagnosis (R05) in this study. This may in part reflect the lack of an appropriate ICD-10 code at the time. In line with this, a survey performed by the European Lung Foundation found that most individuals (>70%) with long lasting cough had visited their specialist physician 3 times or more. Of these, only 53% had received a cough diagnosis [4]. A recently established ICD-10 code specifically for chronic cough will hopefully lead to more individuals receiving an appropriate diagnosis [9].

## Strengths and limitations

The nationwide cohort included a large sample of a well-defined group of patients, which is the main strength of this study. Both the prescription and patient registers are validated on a national level, giving the collected real-world data a high level of credibility and almost 100% coverage [11, 12]. To our knowledge, this is one of very few such studies describing a nationwide population with possible refractory or unexplained chronic cough.

However, there were also some methodological challenges. As inherent to register studies, recorded data was limited by the quality of routinely entered data. As cough is often seen as a symptom rather than a disease, the use of the ICD-10 code R05 for cough is likely underused, especially in RCC. Furthermore, over-the-counter sales of cough medicines is not recorded in the national registers, and thus, chronic cough individuals who only self-medicate will not be captured.

The lack of data from primary care for many Swedish healthcare regions led to a significant number of missing individuals with PCC, as seen by the doubled prevalence in a region with primary care data. However, the characteristics of the patient groups identified with and without primary care data were in most aspects similar.

Identifying RCC/UCC patients based on register data only has some challenges [22]. Some of the information such as adequate treatment of underlying conditions (for RCC) or diagnostic workup process (for UCC) is not available from the register data. Repeated acute cough episodes may have been misclassified as chronic cough, or vice versa. There is a need for improved diagnostic codes characterizing chronic cough to minimize misclassification. Smoking and obesity is also known to be associate with chronic cough [19, 22, 23]. Unfortunately, Swedish registry data does not include BMI or identify smokers, and therefore we could not explore the impact of obesity and smoking on chronic cough. Also, treatment response is not included in the registers, and therefore treatment response could not be assessed.

Nevertheless, the data provide an important presentation of the characteristics of a nationwide population with possible refractory or unexplained chronic cough, and how chronic cough has been managed in Sweden without formal diagnostic or treatment guidelines.

## Conclusions

This nationwide observational register-based study contributes to a better understanding of individuals affected by refractory or unexplained chronic cough. Possible chronic cough (PCC) is more prevalent in females and increases with age with a peak around 40–69 years. The majority of individuals with PCC had frequently dispensed medical prescriptions, and a long history of PCC, although often without receiving a formal cough diagnosis. Pharmacological therapy often included the use of opioids. Most individuals with PCC were treated in primary care, without secondary care specialist referrals. This signifies a need for increased awareness of the condition, better treatments, and improved access to specialist care. Further studies are needed to establish how well register-based cohorts of PCC represent chronic cough individuals in general, and how severely they are affected by their cough, to further increase the understanding of this debilitating condition and improve patient care.

## Supporting information

**S1 Fig. Age and gender distribution of individuals with possible chronic cough, by A) those identified from prescriptions only, B) those identified from diagnosis only, C) those identified with both prescriptions and diagnosis, D) those identified in the regional cohort.**
(TIF)

**S2 Fig. Distribution in the 21 county councils in Sweden of included individuals with chronic cough due to ICD 10 code R05 or frequent dispensed prescriptions of relevant cough medicine.**
(TIF)

**S1 File.**
(DOCX)

## Acknowledgments

We want to thank Prof Eva Millqvist for valuable and important input to the study protocol.

## Author Contributions

**Conceptualization:** Lotta Walz, Össur Ingi Emilsson.

**Data curation:** Kristoffer Illergård, Johannes Arpegård, Cristian Dorbesi.

**Formal analysis:** Kristoffer Illergård, Johannes Arpegård, Cristian Dorbesi.

**Investigation:** Lotta Walz.

**Methodology:** Lotta Walz, Henrik Johansson, Össur Ingi Emilsson.

**Project administration:** Lotta Walz.

**Resources:** Lotta Walz.

**Software:** Cristian Dorbesi.

**Supervision:** Össur Ingi Emilsson.

**Validation:** Kristoffer Illergård, Johannes Arpegård, Henrik Johansson, Össur Ingi Emilsson.

**Visualization:** Cristian Dorbesi, Össur Ingi Emilsson.

**Writing – original draft:** Lotta Walz, Henrik Johansson, Össur Ingi Emilsson.

**Writing – review & editing:** Kristoffer Illergård, Johannes Arpegård, Cristian Dorbesi, Henrik Johansson, Össur Ingi Emilsson.

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
