## [Decision Letter · Decision Letter 0]

24 Jan 2024

PONE-D-23-39518Characteristics, demographics, and epidemiology of possible chronic cough in Sweden: A nationwide register-based cohort studyPLOS ONE

Dear Dr. Walz,

Thank you for submitting your manuscript to PLOS ONE. After careful consideration, we feel that it has merit but does not fully meet PLOS ONE’s publication criteria as it currently stands. Therefore, we invite you to submit a revised version of the manuscript that addresses the points raised during the review process.

We look forward to receiving your revised manuscript.

Kind regards,

Sherief Ghozy, M.D.

Academic Editor

PLOS ONE

A clean copy of the edited manuscript (uploaded as the new *manuscript* file).

“This study was supported by MSD (Sweden) AB. The funding was used by REVEAL for data extractions, and time spent for aggregation and analysis of data (grant/award number: not applicable).”

“We want to thank Prof Eva Millqvist for valuable and important input to the study protocol. Appreciations to MSD (Sweden) AB for the unrestricted funding to support time spent for aggregation and analysis of data.”

“This study was supported by MSD (Sweden) AB. The funding was used by REVEAL for data extractions, and time spent for aggregation and analysis of data (grant/award number: not applicable).”

6. Thank you for stating the following in the Competing Interests section:

“Conflicts of interest:The authors report no conflicts of interest in this work, except that LW is employed by MSD (Sweden) AB, but with no stock ownership. ÖE has received honoraria from MSD and AstraZeneca for advisory work, unrelated to this manuscript.”

7. In this instance it seems there may be acceptable restrictions in place that prevent the public sharing of your minimal data. However, in line with our goal of ensuring long-term data availability to all interested researchers, PLOS’ Data Policy states that authors cannot be the sole named individuals responsible for ensuring data access (http://journals.plos.org/plosone/s/data-availability#loc-acceptable-data-sharing-methods).

8. Your ethics statement should only appear in the Methods section of your manuscript. If your ethics statement is written in any section besides the Methods, please delete it from any other section.

9. We note that Fig S2 in your submission contain [map/satellite] images which may be copyrighted. All PLOS content is published under the Creative Commons Attribution License (CC BY 4.0), which means that the manuscript, images, and Supporting Information files will be freely available online, and any third party is permitted to access, download, copy, distribute, and use these materials in any way, even commercially, with proper attribution. For these reasons, we cannot publish previously copyrighted maps or satellite images created using proprietary data, such as Google software (Google Maps, Street View, and Earth). For more information, see our copyright guidelines: http://journals.plos.org/plosone/s/licenses-and-copyright.

1. You may seek permission from the original copyright holder of Fig S2 to publish the content specifically under the CC BY 4.0 license. 

10. We notice that your supplementary Figures and tables are included in the manuscript file. Please remove them and upload them with the file type 'Supporting Information'. Please ensure that each Supporting Information file has a legend listed in the manuscript after the references list.

Reviewers' comments:

Reviewer's Responses to Questions

**Comments to the Author**

1. Is the manuscript technically sound, and do the data support the conclusions?

Reviewer #1: Partly

Reviewer #2: Yes

2. Has the statistical analysis been performed appropriately and rigorously? 

Reviewer #1: I Don't Know

Reviewer #2: Yes

3. Have the authors made all data underlying the findings in their manuscript fully available?

Reviewer #1: No

Reviewer #2: Yes

4. Is the manuscript presented in an intelligible fashion and written in standard English?

Reviewer #1: Yes

Reviewer #2: Yes

5. Review Comments to the Author

Reviewer #1: Dear Author

The register-based cohort study in a Sweden population is very interesting.

But the data is only about demographics of population with cough.

There is no new or interesting finding related to cough in population in the manuscript.

Reviewer #2: A good study addressing chronic cough.

# Smoking is also one of the major causes for chronic cough even without established COPD. Addressing it becomes mandatory.

# Not only treating but also assessing the treatment response will help physicians across world to provide better patient care. Provide the data for treatment response if available.

# Discussion seems superficial. Dig and include more comparative studies.

6. PLOS authors have the option to publish the peer review history of their article (what does this mean?). If published, this will include your full peer review and any attached files.

Reviewer #1: No

Reviewer #2: No

---

## [Author Response · Author response to Decision Letter 0]

4 Mar 2024

Reviewer #1: Dear Author

The register-based cohort study in a Sweden population is very interesting.

Reply: Thank you for this positive comment!

But the data is only about demographics of population with cough.

There is no new or interesting finding related to cough in population in the manuscript.

Reply: This is an important comment. The cohort described shows a cohort that in most aspects is similar to those found in clinical studies on chronic cough. This is important to show, as this is the first study identifying patients with chronic cough using the nationwide Swedish healthcare register data, indicating that our method finds patients with chronic cough similar to those identified in clinical studies. In that aspect, the findings are novel. We have now emphasized this in the discussion section, line 304-306, by adding the following: “This indicates that the patient group identified with PCC using the Swedish healthcare register data is comparable to chronic cough cohorts recruited in clinical studies.”

Also, the real-world prevalence within the healthcare setting, as well as where the patients are treated, has not been shown before.

Reviewer #2: A good study addressing chronic cough.

# Smoking is also one of the major causes for chronic cough even without established COPD. Addressing it becomes mandatory.

Reply: This is an important comment. Unfortunately, the Swedish healthcare register data does not include any data on smoking, and therefore this important factor could not be included in the analysis. This is addressed in the discussion, section on limitations, but we have now also added this to the methods section, lines 145-146: “As the registers do not have information on smoking, smoking status could not be used in the exclusion criteria.”

# Not only treating but also assessing the treatment response will help physicians across world to provide better patient care. Provide the data for treatment response if available.

Reply: Unfortunately, treatment response is also not included in the Swedish healthcare register data, and could therefore not be included. This is now highlighted in the section on limitations, lines 347-349: “Also, treatment response is not included in the registers, and therefore treatment response could not be assessed.”

# Discussion seems superficial. Dig and include more comparative studies.

Reply: Thank you for this suggestion. We have now made some adjustments: In response to a comment made by reviewer 1, we have emphasized the novelty of the method used (see reply above). Also, we have added a deeper interpretation of our finding regarding patients being largely treated in primary care only by adding the following in lines 272-274: “A recent international Delphi study identified variability between geographical regions regarding the treatment of chronic cough, and a need for increased access to specialist care to improve patient care.” with reference to “Song WJ et al, 2023, ERJ Open Research”. Finally, we added to the discussion on opioid use in chronic cough, lines 312-313: “This is a somewhat higher proportion of opioid use compared with recent data from a study in Florida.” with reference to “Yang S et al, 2023, J Clin Med”.

---

## [Editor Report · Decision Letter 1]

1 May 2024

Characteristics, demographics, and epidemiology of possible chronic cough in Sweden: A nationwide register-based cohort study

PONE-D-23-39518R1

Dear Dr. Walz,

We’re pleased to inform you that your manuscript has been judged scientifically suitable for publication and will be formally accepted for publication once it meets all outstanding technical requirements.

Kind regards,

Sherief Ghozy, M.D.

Academic Editor

PLOS ONE

---

## [Editor Report · Acceptance letter]

15 May 2024

PONE-D-23-39518R1 

PLOS ONE

Dear Dr. Walz, 

I'm pleased to inform you that your manuscript has been deemed suitable for publication in PLOS ONE. Congratulations! Your manuscript is now being handed over to our production team.

Kind regards, 

on behalf of

Dr. Sherief Ghozy 

Academic Editor

PLOS ONE